# Optimization of Sparse Sensor Layouts and Data-Driven Reconstruction Methods for Steady-State and Transient Thermal Field Inverse Problems

**DOI:** 10.3390/s25164984

**Published:** 2025-08-12

**Authors:** Qingyang Yuan, Peijun Yao, Wenjun Zhao, Bo Zhang

**Affiliations:** 1Key Laboratory of Complex Energy Conversion and Efficient Utilization of Liaoning Province, School of Energy and Power Engineering, Dalian University of Technology, Dalian 116081, China; yuan_qingy@mail.dlut.edu.cn (Q.Y.); zwj1218@mail.dlut.edu.cn (W.Z.); 2Zhejiang SHIP Electronics Technology Co., Ltd., Ningbo 315191, China; pepi1@shipgroup.net; 3Ningbo Research Institute, Dalian University of Technology, Ningbo 315032, China

**Keywords:** inverse temperature field reconstruction, sparse sensor layout optimization, gappy C-POD, data-driven reconstruction, steady-state and transient heat conduction

## Abstract

This paper investigates the inverse reconstruction of temperature fields under both steady-state and transient heat conduction scenarios. The central contribution lies in the structured development and validation of the Gappy Clustering-based Proper Orthogonal Decomposition (Gappy C-POD) method—an approach that, despite its conceptual origin alongside the clustering-based dimensionality reduction method guided by POD structures (C-POD), had previously lacked an explicit algorithmic framework or experimental validation. To this end, the study constructs a comprehensive solution framework that integrates sparse sensor layout optimization with data-driven field reconstruction techniques. Numerical models incorporating multiple internal heat sources and heterogeneous boundary conditions are solved using the finite difference method. Multiple sensor layout strategies—including random selection, S-OPT, the Correlation Coefficient Filtering Method (CCFM), and uniform sampling—are evaluated in conjunction with database generation techniques such as Latin Hypercube sampling, Sobol sequences, and maximum–minimum distance sampling. The experimental results demonstrate that both Gappy POD and Gappy C-POD exhibit strong robustness in low-modal scenarios (1–5 modes), with Gappy C-POD—when combined with the CCFM and maximum distance sampling—achieving the best reconstruction stability. In contrast, while POD-MLP and POD-RBF perform well at higher modal numbers (>10), they show increased sensitivity to sensor configuration and sample size. This research not only introduces the first complete implementation of the Gappy C-POD methodology but also provides a systematic evaluation of reconstruction performance across diverse sensor placement strategies and reconstruction algorithms. The results offer novel methodological insights into the integration of data-driven modeling and sensor network design for solving inverse temperature field problems in complex thermal environments.

## 1. Introduction

Heat conduction problems are prevalent across various engineering applications, including mechanical engineering, architectural engineering, and the thermal management of electronic devices. The inverse problem of temperature fields [1,2], which involves deducing the distribution of heat sources and boundary conditions from known temperature data, is a critical technique for addressing these challenges. Traditional numerical simulation methods, particularly those based on finite difference methods (FDMs) [3] and finite element methods (FEMs) [4], often rely on comprehensive sensor networks and detailed boundary conditions. However, in practical scenarios, sensor deployment frequently suffers from sparsity, making it difficult to accurately capture the complete temperature field information, thereby complicating the solution of the temperature field inverse problem.

Classical methods for solving thermal inverse problems primarily encompass analytical methods [5], regularization techniques [6], optimization strategies [7], Bayesian approaches [8], and iterative inversion methods [9]. Analytical methods yield precise solutions through mathematical derivation; in the fields of fluid flow and heat transfer, Fourier transforms, Green’s functions, and Laplace inversion [10] are widely utilized. These methods typically apply to linear problems with simple models and regular boundary conditions [11,12]. For instance, in steady-state conduction problems where thermal conductivity varies linearly with temperature, inverse analysis can be employed for parameter estimation. Although analytical solutions can offer accurate results with clear physical interpretations, they are mainly constrained by stringent requirements regarding the linearity of the problem and geometric regularity, making them inadequate for complex nonlinear problems and irregular geometries.

Ill-posedness is an inherent challenge of inverse problems, where even minor data noise can lead to significant deviations in solutions. Regularization methods aim to stabilize solutions by introducing additional constraints, thereby enhancing robustness. Tikhonov regularization is one of the most established techniques, which smooths solutions by adding a penalty term to the least-squares objective function, achieving a balance between the data fitting error and the “smoothness” or “simplicity” of the solution [13]. In the context of heat transfer, regularization methods are extensively employed to identify unknown parameters within mathematical models. For instance, when dealing with ill-posed problems, various regularization techniques can be selected with appropriately adjusted regularization parameters [14]. However, the selection of these parameters often depends on empirical judgment, and inappropriate choices may lead to erroneous results.

Optimization methods convert inverse problems into optimization tasks, estimating unknown parameters by minimizing the residuals between observed data and model predictions. Prominent approaches include nonlinear least squares, constrained optimization, and variational methods [15]. These methods are versatile and theoretically sound, finding broad applications in parameter estimation, imaging, and system identification. For example, in intricate heat transfer scenarios, they can optimize thermal performance designs [16]. Nevertheless, these methods are susceptible to local optima and may struggle with multimodal problems, while computational efficiency is restricted by the problem’s scale and complexity [17]. Bayesian methods, rooted in probability theory, treat unknown parameters as random variables, updating posterior probability distributions by synthesizing prior knowledge with observational data to quantify uncertainty. Techniques such as Markov Chain Monte Carlo (MCMC) and variational Bayes are commonly employed [18]. This approach excels in managing complex models and quantifying uncertainties, as exemplified by predicting the energy consumption of HVAC systems using Bayesian networks [19]. The primary challenges arise from the computational demands of high-dimensional parameter spaces and the intricacies involved in model construction. Iterative inversion methods approach the true solution through successive iterations, with prevalent algorithms including Landweber iteration, the Conjugate Gradient Method (CGM), and Gauss–Newton methods [20]. These techniques are straightforward to implement, capable of addressing nonlinear problems, and are widely used in image reconstruction, bioelectrical impedance tomography (BIT), and seismic imaging. For example, when solving ill-posed linear inverse problems, a recently proposed second-order dynamic method (SODM) integrates Tikhonov regularization with second-order asymptotic regularization and employs a dual-parameter strategy, demonstrating superiority over conventional methods under noisy conditions [21]. However, it is essential to note that this approach has slower convergence rates, sensitivity to initial values, and demands meticulous parameter tuning [22].

With the advancement of machine learning and data-driven methodologies, data-based field reconstruction techniques have increasingly emerged as effective tools for addressing related challenges. These approaches effectively reconstruct accurate temperature field distributions by leveraging limited sensor data, integrated with dimensionality reduction techniques and model simplification strategies. Specifically, the Proper Orthogonal Decomposition (POD) method [23] facilitates dimensionality reduction in temperature field data, extracting principal feature modes and thereby simplifying complex heat transfer problems. The Gappy POD method [24] is an extension of POD designed to handle incomplete data. In heat transfer scenarios, complete temperature field data may not be obtainable due to sensor malfunctions or data transmission errors. Gappy POD allows for the reconstruction of the full temperature field from available partial data. For example, research conducted by Wang et al. [25] involved deploying a limited number of sensors, combined with Computational Fluid Dynamics (CFD) simulations and the Gappy POD methodology, to reconstruct the temperature field distribution across an entire indoor space. Moreover, Xiao et al. [26] addressed the severe missing data issues in meteorological observations by generating a POD basis using the WRF model and subsequently employing Gappy POD to fill in the gaps, successfully reconstructing the temperature field in the Tibetan region. The POD-RBF (Proper Orthogonal Decomposition–Radial Basis Function) method effectively reconstructs temperature fields with a small number of measurement points. This technique utilizes POD to extract key characteristics of the temperature field and employs RBF interpolation for complete temperature reconstruction, thus striking a balance between computational efficiency and accuracy. Research by Wang et al. [27] demonstrated the use of the POD-RBF method to rapidly reconstruct the temperature field of printed circuit boards (PCBs) based on minimal sensor data, facilitating monitoring and control of thermal behavior in devices. The Multilayer Perceptron (MLP) represents a type of artificial neural network capable of learning nonlinear relationships between inputs and outputs. The POD-MLP method integrates POD dimensionality reduction with MLP neural networks, aiding in the estimation of physical system states, including thermal and flow conditions. Qi et al. [28] leveraged the POD-MLP technique for real-time monitoring and reconstruction of the surface temperature field of batteries. The accuracy of data-driven field reconstruction methods significantly depends on the sparse sensor layout. A strategically designed sensor arrangement can not only enhance the precision of temperature field reconstruction but also effectively reduce data acquisition costs and computational burdens. However, many existing reconstruction methods overlook the influence of sparse sensor configurations, often resorting to uniform or random placement techniques, which can lead to considerable variability in reconstruction accuracy. Willcox et al. [29] optimized the sparse sensor placement for the Gappy POD method using matrix stability criteria, specifically condition numbers, achieving commendable results; however, this approach demands iterative and repeated matrix computations, resulting in lower time efficiency. It also necessitates integration with optimization methods to accelerate the selection process in complex scenarios. Lauzon et al. [30] introduced the S-OPT method, which quickly determines optimal sensor configurations by scanning orthogonal basis matrices to maximize an S-metric. Yet, this method proves sensitive to the dimensionality of the basis, requiring higher-order modes for accurate results in complex fields such as turbulence. Yuan et al. [31] proposed a method utilizing the Pearson correlation coefficients of matrices to gauge the interrelationships between sensors and employed a threshold-based strategy to swiftly identify optimal sensor placement locations, known as the Correlation Coefficient Filtering Method (CCFM). This method enables sparse sensor layout optimization under low-modal and confined regional conditions.

Building upon the clustering-based dimensionality reduction method guided by pod structures proposed (C-POD) by Yuan et al. [31], this paper introduces the Gappy C-POD approach, incorporating principles from inverse problem solving, and develops a multi-scale sample database alongside various sensor layout optimization strategies to facilitate its implementation. This study systematically compares the reconstruction pathways, accuracy, and robustness of four data-driven reconstruction methods—Gappy POD, Gappy C-POD, POD-RBF, and POD-MLP—across different modal truncation numbers, database sizes, and selection strategies, focusing on steady-state and transient heat conduction problems with multiple internal heat sources in two dimensions. The findings aim to provide theoretical foundations and methodological guidance for the engineering modeling and optimization design of temperature field inverse problems.

## 2. Overview of Gappy C-POD and Data-Driven Field Reconstruction Methods

### 2.1. Introduction to POD and Gappy POD Methods

Proper Orthogonal Decomposition (POD), introduced by Sirovich et al. [32], is a well-established technique for dimensionality reduction and feature extraction. It has become widely adopted for modal decomposition and data compression of high-dimensional physical fields. The POD method decomposes fields, such as temperature distributions, to identify modal bases that encapsulate where energy is predominantly concentrated, thereby characterizing the primary structures and dynamics of the field. For a snapshot matrix of the temperature field containing Ns samples $T=[T_{1},T_{2},\ldots ,T_{N_{S}}]$, its covariance matrix R can be expressed as follows:(1)$$R=\frac{1}{N_{S}}TT^{T}$$

Next, by addressing the eigenvalue problem presented in the subsequent Equation (2):(2)$$R\phi_{i}=\lambda_{i}\phi_{i}$$
one can derive the basis functions ***ϕ****_i_* along with their corresponding energy ***λ****_i_*. The temperature field can be expanded in terms of these modes as follows:(3)$$T\approx \sum_{i=1}^{r}a_{i}\phi_{i}=\phi_{r}a$$

In this expression, the modal basis matrix $\phi_{r}=[\phi_{1},\ldots ,\phi_{r}]$ retains the first r modes, with ***a*** being the modal coefficient vector. In practical scenarios, due to limitations in data acquisition, temperature observations are often sparse or incomplete. The Gappy POD method was developed to address this issue, aiming to reconstruct the complete field under sparse observation conditions. Let us denote the sparse observed temperatures as $T_{obs}$ with the observation matrix being ***C*** (which extracts the relevant rows corresponding to the observed points); the objective of Gappy POD is to minimize the residuals at the observed points as follows:(4)$$\underset{a}{\min}{\left\|C\phi_{r}a-T_{obs}\right\|}^{2}$$

The least-squares solution can be computed using(5)$$a={\left(C\phi_{r}\right)}^{+}T_{obs}$$

Here, the operation ${\left(\cdot \right)}^{+}$ signifies the generalized inverse (pseudo-inverse). Consequently, the complete reconstructed temperature field takes the form of(6)$$T_{\mathrm{recon}}=\phi_{r}a$$

The Gappy POD method demonstrates robust reconstruction capabilities even in cases of data sparsity or local deficiencies. The accuracy of the reconstruction is influenced by the energy distribution of the modal basis, the configuration of the observation points, and the numerical condition of the observation matrix. Therefore, an effective strategy for optimizing the layout of sparse sensors is crucial to ensuring the optimal performance of the Gappy POD approach.

### 2.2. Introduction to C-POD and Gappy C-POD Methods

The linear assumptions underlying traditional POD constrain its effectiveness in reconstructing fields characterized by strong nonlinearity. To overcome this limitation, Yuan et al. [31] introduced a clustering-based dimensionality reduction method guided by POD structures (C-POD), which combines POD modal structures with clustering algorithms to enhance local feature extraction. Although the conceptual notion of a Gappy C-POD—a gappy extension of C-POD to handle sparse observation scenarios—was proposed concurrently, its complete technical implementation and empirical validation have not been previously reported. In this study, we present the first structured implementation of the Gappy C-POD method, integrating clustering-based modal decomposition with sparse sensor data to reconstruct temperature fields under both steady-state and transient heat conduction conditions. This approach improves local adaptability while maintaining computational efficiency, and shows particular advantages under low-mode and sensor-limited settings.

Initially, C-POD utilizes the low-dimensional modal basis generated by POD as the starting point for clustering, selecting the first K bases as the initial cluster centers $\mu_{i}^{\left(0\right)}$:(7)$$\mu_{i}^{\left(0\right)}=\phi_{i},i=1,2,\ldots ,K$$

The iterative update of these clustering centers can be expressed by(8)$$\mu_{i}^{\left(t\right)}=\frac{1}{\left|S_{i}^{\left(t\right)}\right|}\sum_{T_{j}\in S_{i}^{\left(t\right)}}T_{j}$$

Here, the notation $S_{i}^{\left(t\right)}$ represents the collection of temperature samples belonging to the i-th cluster at the t-th iteration, while $T_{j}$ denotes the j-th temperature field sample. By projecting each group of temperature field samples onto the clustering centers, we derive the modal coefficients associated with the C-POD method, articulated as follows:(9)$$a_{j}={\left(\mu^{T}\mu \right)}^{-1}\mu^{T}T_{j}$$

Given the observation matrix ***C*** that extracts sparse temperature observations, the observed temperature data can be formulated as(10)$$T_{obs}=CT$$

This leads to the modal coefficient Equations (11) and (12) under sparse observation conditions:(11)$$a=\arg\underset{a}{\min}{\left\|C\mu a-T_{obs}\right\|}^{2}$$(12)$$a={\left(C\mu \right)}^{+}T_{obs}$$

The resulting reconstructed temperature field can be expressed as(13)$$\hat{T}=\mu a$$

The clustered modal basis ***μ*** in C-POD encapsulates the local nonlinear characteristics inherent in the data, effectively overcoming the limitations of the linear orthogonal basis expansion employed by traditional POD. The Gappy C-POD method further enhances this by capturing local features and the multi-state evolution of nonlinear flow fields under conditions of sparse observation.

### 2.3. Solutions to Inverse Problems Using POD-RBF

In the context of data-driven inverse problem solving, the POD-RBF method establishes a nonlinear mapping from sparse sensor observations to POD modal coefficients or directly to full temperature fields. This approach effectively overcomes the limitations of conventional linear least-squares POD reconstruction techniques, particularly in scenarios involving nonlinear dynamics, irregular boundaries, or strong disturbances.

This nonlinear mapping is implemented using a Radial Basis Function (RBF) neural network, which is well-known for its structural simplicity, fast convergence, and strong adaptability to small-sample datasets. In the POD-RBF framework, the reconstruction process begins with extracting POD modal coefficients ***a*** from the training data using Proper Orthogonal Decomposition. These coefficients are then regressed against sparse observations via the trained RBF network, enabling efficient and flexible reconstruction even under sparse and noisy input conditions.(14)$$T\approx \phi_{r}a$$

Following this, a Radial Basis Function (RBF) network is established to create a nonlinear mapping between the sparse observational data ***T***_obs_ and the modal coefficients ***a***:(15)$$a=\sum_{i=1}^{N_{c}}\omega_{i}\varphi (\left\|T_{obs}-C_{i}\right\|)$$

In this equation, φ(⋅) stands for the RBF kernel function; $C_{i}$ signifies the centers; $\omega_{i}$ represents the weight parameters; and *N_c_* indicates the number of centers. The reconstructed temperature field is detailed as follows:(16)$$\hat{T}=\phi_{r}a$$

### 2.4. Solutions to Inverse Problems Using POD-MLP Methods

In comparison to RBF methods, Multilayer Perceptron (MLP) neural networks exhibit superior fitting capability and are more suitable for larger sample datasets. Using the sparse observational data ***T***_obs_ as the input, a Multilayer Perceptron (MLP) neural network is trained:(17)$$a=f_{MLP}(T_{obs};\theta )$$

In this expression, $f_{MLP}$ refers to the MLP network function, while θ denotes the network weights and bias parameters, respectively. The MLP network, through multiple layers of nonlinear activation units, approximates the complex mapping relationship between observation points and modal coefficients. Upon the completion of its training, it can swiftly predict modal coefficients and reconstruct the complete field.

## 3. Reconstruction of Steady-State Temperature Field

### 3.1. Case Study Overview

To evaluate the applicability and robustness of the methodology for solving inverse problems related to temperature fields, this study presents a classic example of a two-dimensional steady-state heat conduction problem, as illustrated in Figure 1. The model simulates the heat transfer process influenced by multiple irregularly distributed heat sources and certain isothermal boundary conditions, employing the finite difference method for numerical analysis.

#### 3.1.1. Geometric Parameters and Heat Source Distribution

As depicted in Figure 1, the computational domain in this case has a side length of L = 0.1 m. Within this domain, there are ten rectangular heat sources, the locations, dimensions, and intensity of which are specified in Table 1:

#### 3.1.2. Governing Equation

The governing equation is a nonlinear steady-state heat conduction equation, represented as shown in Equation (18):(18)∇⋅(λ(T)∇T)+Θ(x,y)=0,(x,y)∈Ω
where Ω=[0,L]×[0,L] is defined as a two-dimensional square area, λ(T)=1+0.05(T−298), and Θ(x,y) is the heat source term is included.

#### 3.1.3. Initial Conditions

The initial temperature field is uniformly distributed, as expressed in Equation (19):(19)T(x,y)=298,(x,y)∈Ω

#### 3.1.4. Boundary Conditions

The boundary conditions are specified as mixed. The left and right boundaries are characterized by adiabatic conditions:(20)$$\frac{\partial T}{\partial x}=0$$

The upper boundary also maintains an adiabatic condition:(21)$$\frac{\partial T}{\partial y}=0$$

For the lower boundary, most of it is adiabatic, except for the central position, which is kept at a constant temperature, as illustrated in Equation (22):(22)$$T=298K,x\in [\frac{L}{2}-0.005,\frac{L}{2}+0.005]$$

#### 3.1.5. Solution Method

The solution is obtained through the finite difference method:Grid: A uniform rectangular grid of 100 × 100 is employed.Spatial Discretization: A nonlinear five-point difference scheme is used.Nonlinear Solving: The Gauss–Seidel iterative method is applied as per Equation (23):(23)$$T_{i,j}=\frac{\lambda_{E}T_{i,j+1}+\lambda_{W}T_{i,j-1}+\lambda_{N}T_{i-1,j}+\lambda_{S}T_{i+1,j}+\Delta x^{2}\Theta_{i,j}}{\lambda_{E}+\lambda_{W}+\lambda_{N}+\lambda S}$$
where $\lambda_{E}=1+0.05(T_{i,j+1}-298)$ and the representation of other directions is analogous. The convergence criterion is established in Equation (24), with the maximum iteration limit set to 10,000.(24)$$\max\left|T^{\left(n+1\right)}-T^{\left(n\right)}\right|<10^{-5}K$$

### 3.2. Construction of Verification Set

As indicated in Section 2.1, the thermal conduction problem incorporates ten internal heat sources as independent variables. A test set comprising 200 randomly sampled groups was created, with the distribution depicted in Figure 2. The boxplot illustrates the thermal power values for each heat source, uniformly spanning the defined range of (0.5–2.0) × 10^5^ W, devoid of outliers. This indicates sufficient input coverage for both model training and validation purposes.

### 3.3. Database Construction

To establish a representative and widely encompassing sample database, this study employs three typical sampling methods: Latin Hypercube sampling [33], Sobol sequence sampling [34], and maximum distance sampling [35]. These methods are applied to sample the combination space of ten thermal power variables. Given the dimensionality of the variables, denoted as *d* = 10, sample sizes were set at five, ten, twenty, thirty, forty, and fifty times the number of variables, resulting in the generation of 50, 100, 200, 300, 400, and 500 sets of independent variable combinations. This yields a total of 18 multi-scale sample databases that will be utilized for subsequent modeling and analysis.

### 3.4. Sparse Sensor Layout Optimization Methods

To investigate the impact of different sparse sensor layout optimization methods on field reconstruction performance, four representative sensor placement strategies were selected: random sampling, the S-OPT method, the Correlation Coefficient Filtering Method (CCFM), and uniform sampling. For each strategy, the number of sensors was varied at 4, 9, 16, 25, 36, and 49, resulting in a total of 24 distinct sparse sensor layout configurations, as illustrated in Figure 3.

### 3.5. Reconstruction Methods

To evaluate the reconstruction capabilities of various methods, this study compares four representative approaches: Gappy Proper Orthogonal Decomposition (POD), Gappy Composite Proper Orthogonal Decomposition (C-POD), POD–Radial Basis Function (RBF), and POD–Multilayer Perceptron (MLP). Under the conditions specified in items (3) to (5), a systematic investigation was conducted based on the validation set detailed in Section 2. This research focused on the impact of various factors on reconstruction accuracy, including different dimensional reduction modes, database construction methods, sparse sensor layout optimization strategies, and reconstruction methods. To quantify the performance of each reconstruction approach concerning maximum deviation constraints and overall accuracy, the study employs two key metrics—the minimum maximum relative error (MRE) and the minimum average relative error (MARE)—defined mathematically by Equations (25) and (26):(25)$$\mathrm{MRE}=min_{j=1}^{N}\left(\max_{i=1}^{n}\left|\frac{T_{i,j}^{rec}-T_{i}^{true}}{T_{i}^{true}}\right|\right)$$(26)$$\mathrm{MARE}=min_{j=1}^{N}\left(\frac{1}{n}\sum_{i=1}^{n}\left|\frac{T_{i,j}^{rec}-T_{i}^{true}}{T_{i}^{true}}\right|\right)$$
where $T_{i,j}^{rec}$ is the reconstructed temperature at the *i*-th point under the *j*-th reconstruction configuration; $T_{i}^{true}$ is the true (reference) temperature at the *i*-th point; *n* is the total number of temperature points; and *N* is the total number of reconstruction configurations.

The reconstruction approach is denoted as x_y_z_m, where x indicates the reconstruction method (Gappy POD, Gappy C-POD, POD-RBF, POD-MLP), y refers to the database size (5, 10, 20, 30, 40, and 50 times the number of independent variables), z denotes the point selection strategy (CCFM, S-OPT, Random, Uniform), and m indicates the number of selected points (4, 5, 16, 25, 36, 49).

### 3.6. Analysis of Optimal Reconstruction Approaches for Different Modal Counts

To analyze the performance variations in different reconstruction methods within varying modal ranges, this study collects data on the minimum maximum relative error (Min(MRE)) and minimum average relative error (Min(MARE)) for modes ranging from 1 to 5, 5 to 10, 10 to 15, and 15 to 20. A smaller Min(MRE) indicates a superior capability of the reconstruction scheme in limiting reconstruction deviation within the corresponding mode range (i.e., higher stability), while a smaller Min(MARE) reflects greater overall reconstruction accuracy. The results indicate that the Min(MRE) for all methods does not decrease with an increase in modal count. Notably, at the 15th modal count, all methods achieved the strongest capacity for limiting reconstruction deviation. Table 2 presents the optimal implementation paths and corresponding error values for each reconstruction method when reaching Min(MRE) within the 15-mode range. It is evident from the table that the approach Gappy C-POD_Maximum_50_CCFM_49 demonstrates the most effective deviation control, suggesting that this scheme is less prone to significant errors during the reconstruction process.

Further observations reveal that, under the condition of achieving minimal reconstruction error, the optimal point selection strategy employed by all reconstruction methods is the Correlation Coefficient Filtering Method (CCFM). An analysis combining Figure 1 and Figure 3 suggests that the CCFM exhibits a higher concentration of selected points in the heat source distribution area, thereby enhancing its ability to capture critical physical features and improving overall reconstruction quality. Additionally, in scenarios with a limited number of modes, the POD-MLP method effectively suppresses deviations even when relying on a small-scale database (20 times the number of independent variables) and a minimal number of sensor points (nine). This finding further indicates that utilizing maximum distance sampling and Sobol sequence sampling for database construction under these conditions yields superior results.

Figure 4 illustrates the distribution of absolute reconstruction errors across various methods on the validation set when the Gappy C-POD_Maximum_50_CCFM_49 approach achieves Min(MRE). It is evident from the figure that the maximum reconstruction error of the Gappy C-POD method is less than 0.72 °C, significantly outperforming the other three reconstruction methods, thereby reaffirming its superiority in low-order modes.

The temperature field modes obtained through the dimensionality reduction techniques of POD and C-POD are illustrated in Figure 5. This figure displays the characteristic modal structures extracted from temperature field samples using both POD and C-POD methodologies. The upper section of the figure presents the first to nth modes extracted via the POD approach, while the lower section showcases the corresponding clustered modes generated by the C-POD method.

From the observations in Figure 5, the following conclusions can be drawn:(1)The POD modes (upper section) exhibit commendable spatial smoothness and energy concentration, with the lower-order modes primarily reflecting the temperature gradient variations aligned with the main direction of the heat sources, thereby encapsulating the overall characteristics of the temperature field. In contrast, the higher-order modes contain more detailed local perturbation information, although their physical interpretability is comparatively limited.(2)The C-POD modes (lower section) display a greater complexity in spatial structure, highlighting pronounced local features that enable precise focus on areas of intense heat source concentration or significant temperature fluctuations. Particularly under complex boundary conditions or irregular heat source distributions, the C-POD modes demonstrate enhanced regional adaptability, with modal shapes more closely resembling the nonlinear distribution of actual physical fields.

From the perspective of modal structural characteristics, the POD approach is more effective for extracting dominant global features, benefitting from strong mathematical orthogonality and energy spectral decay properties. Conversely, the C-POD modes exhibit superior capabilities in local feature extraction and nonlinear pattern recognition. This distinction is further corroborated in subsequent comparative analyses, which indicate that under conditions of low modal counts, the C-POD method significantly outperforms traditional POD in terms of reconstruction error management and stability, making it particularly suited for the reconstruction of non-uniform temperature fields in scenarios with sparse sensor configurations.

The trend of Min(MARE) values for different reconstruction methods as the number of modes increases is illustrated in Figure 6. It is evident from this figure that the Min(MARE) values decrease progressively with an increasing number of modes. The overall reconstruction accuracy can be ranked from lowest to highest as follows: POD-RBF, Gappy C-POD, POD-MLP, and Gappy POD. Notably, both the Gappy POD and POD-MLP methods demonstrate superior overall reconstruction accuracy.

Table 3 summarizes the pathways through which the various methods achieve their Min(MARE) values when the number of modes is between 15 and 20. According to the table, the Gappy POD_Maximum_50_Uniform_9 approach attains the optimal overall reconstruction accuracy. However, it is important to note that the datasets for the Gappy POD, POD-RBF, and POD-MLP methods are considerably larger, each reaching a scale of 50 times the number of independent variables. In contrast, the Gappy C-POD method achieves high precision in field reconstruction with a dataset only five times the size of the independent variables and relies on 25 sparse sensors, making it more suitable for engineering applications. Furthermore, while the machine learning-based POD-MLP and POD-RBF methods show potential in terms of accuracy, their performance is heavily reliant on the quality and size of the dataset. The stability and generalization capabilities of these models still require further validation in scenarios with limited samples.

### 3.7. An Analysis of the Impact of Database Size on the Reconstruction Capability of the Temperature Field Inverse Problem

This section investigates the effect of database size on the performance of inverse problem reconstruction by treating “database size” as the sole independent variable. Under varying sample sizes, a comprehensive consideration is given to all modal truncation numbers and diverse sparse sensor layout strategies (including Gappy POD, Gappy C-POD, POD-MLP, and POD-RBF), as well as multiple configurations of selected point quantities. The optimal combination for each database size will be identified to represent its superior reconstruction capability.

Figure 7 and Figure 8 illustrate the variations in the minimum mean relative error (Min MRE) and the minimum mean absolute relative error (Min MARE) across different database sizes. The data reveal a significant trend of performance enhancement among various reconstruction methods with optimal configurations as the database size increases. Notably, the POD-RBF method demonstrates a remarkable improvement, with its optimal MARE decreasing from 0.00939 at 50 samples to 0.00508 at 500 samples, underscoring the strong dependence of the RBF kernel function model on sample density. Conversely, both Gappy POD and Gappy C-POD reach a performance plateau within a relatively moderate database size range of approximately 200 to 300 samples, indicating efficient data utilization. The POD-MLP method, constrained by its requirements for network parameter convergence, exhibits instability under smaller sample sizes but tends to stabilize when the sample size increases to 500.

## 4. Discussion of Engineering Practice

### Impact of Sparse Sensor Quantity

In engineering applications, the scale of the database and the number of reduced-order modes can significantly influence the overall computational efficiency and deployment capabilities of algorithms. This study examines the effects of database construction methods, the quantity of sparse sensors, and the optimization of sparse sensor layouts on the accuracy of field reconstruction, considering a database scale of five times the number of independent variables and a mode number ranging from 1 to 5. Using four different field reconstruction methods, we reconstructed the test set to obtain Min(MRE) and Min(MARE) values, which vary with the quantity of sparse sensors, as presented in Table 4. For clearer visualization, the data from Table 4 is illustrated in Figure 9 and Figure 10.

Figure 9 indicates that the relationship between Min(MARE) values and sensor quantity is relatively weak. Under conditions of low numbers of modes and smaller datasets, the overall reconstruction accuracy of the four methods is minimally affected by the number of sensors. In this scenario, the methods are ranked from highest to lowest reconstruction accuracy as follows: Gappy POD, POD-MLP, Gappy C-POD, and POD-RBF. Figure 10 shows that the Min(MRE) values for the Gappy POD, Gappy C-POD, and POD-MLP methods demonstrate a decreasing trend in fluctuation as the number of sensors increases. This suggests that augmenting the number of sensors enhances the stability of these three reconstruction methods. Conversely, the Min(MRE) value for the POD-RBF method increases with the addition of sensors. When the number of sensors reaches or exceeds 25, Gappy C-POD consistently exhibits the lowest Min(MRE) value, indicating its strong capacity to constrain maximum reconstruction error and, thus, its enhanced stability under conditions of low modal counts and small sample sizes.

Table 5 and Table 6 summarize the database sampling methods and sensor layout optimization strategies corresponding to the minimum values of max relative error (Min(MRE)) and mean absolute relative error (Min(MARE)) from Table 3. This organization facilitates a clearer comparison of the optimal implementation pathways for various reconstruction methods across differing numbers of sensors. From Table 5, it can be observed that under the condition of minimizing the maximum relative error (Min(MRE)), the Gappy C-POD method achieves optimal performance by employing maximum distance sampling combined with the CCFM (Correlation Coefficient Filtering Method) selection strategy when using 49 sensors. Additionally, both the Gappy POD and POD-MLP methods exhibit a preference for Latin Hypercube sampling and the S-OPT layout across varying sensor counts. In contrast, the POD-RBF method shows a greater reliance on the CCFM selection strategy under either Latin or maximum sampling conditions.

Table 6 reveals that, under the minimization of the mean relative error (Min(MARE)), the Gappy POD method demonstrates a stable adaptability to Sobol sampling and uniform layouts across multiple sensor configurations, particularly exhibiting the highest overall reconstruction accuracy with 49 sensors. Conversely, the Gappy C-POD method maintains relatively low errors at lower sensor counts (such as 4 and 9) while primarily depending on Sobol sampling and either S-OPT or uniform selection approaches. It is noteworthy that the POD-MLP method generally results in lower overall errors but shows a heightened sensitivity to layout methods, particularly relying on maximum sampling and S-OPT layouts. The performance of the POD-RBF method in the Min(MARE) metric is comparatively weaker, especially as the number of sensors increases, where the decrease in error is limited.

In conclusion, the comparison of optimal strategies under conditions of minimum maximum error and minimum average error indicates marked differences in the pathways to achieving optimal performance across various reconstruction methods. The synergy between selection strategies and database construction significantly influences model performance. Notably, the Gappy C-POD method demonstrates greater stability in controlling reconstruction deviation, while the Gappy POD and POD-MLP methods exhibit superior performance in terms of overall accuracy.

## 5. Reconstruction of Transient Thermal Conductivity Temperature Field

### 5.1. Overview of Transient Thermal Conductivity Issue

In examining the transient heat conduction process, we maintain consistency with the boundary conditions and heat source distribution outlined in Section 2.1. The diffusion of heat over time is simulated using an explicit finite difference method. This study investigates the impact of various reconstruction approaches on the reconstruction capacity of the transient temperature field under the influence of non-uniform internal heat sources.

(1)The governing equation for transient heat conduction is represented in Equation (27).(27)$$\frac{\partial T(x,y,t)}{\partial t}=\alpha (\frac{\partial^{2}T}{\partial x^{2}}+\frac{\partial^{2}T}{\partial y^{2}})+\frac{\Theta (x,y)}{\rho c_{p}}$$
where α denotes the thermal diffusivity while Θ(x,y) signifies the heat source density per unit volume.

(2)Materials and numerical parameters.

Materials and numerical parameters, as shown in Table 7, are used in the simulation process. Figure 11 illustrates the evolution of the transient thermal field at various time steps. As time progresses, heat gradually disperses from multiple internal heat sources to the surrounding areas. During the initial phase, localized temperature peaks rapidly emerge around the heat sources. Subsequently, influenced by thermal conduction and boundary conditions, the thermal energy continues to diffuse outward. The temperature field exhibits characteristic features of unsteady heat transfer throughout its evolution, with the thermal structure adjusting continuously over time until it achieves stability.

### 5.2. The Influence of Modal Coefficients on the Reconstruction Capability of Transient Thermal Inverse Problems

In the process of constructing the transient thermal dataset and validation set, the combination of independent variables was maintained as consistent with conditions observed under steady-state heat conduction. For each set of these independent variable combinations, temperature field snapshots were collected every 10 s and incorporated into the dataset. Consequently, in contrast to steady-state thermal scenarios, each sample in the transient heat conduction problem encapsulates multiple time points of temperature field evolution, thereby resulting in datasets and validation sets with matrix dimensions approximately ten times larger than those for steady-state conditions. Apart from the temporal dimension expansion, all other simulation conditions and parameter settings were kept consistent with those in the steady-state heat conduction process to ensure comparability between the two problem types.

To investigate the impact of varying modes on the temperature field reconstruction performance within transient thermal inverse problems, four reconstruction methods (Gappy POD, Gappy C-POD, POD-MLP, and POD-RBF) were employed under conditions of sparse sensor data. The performance of these methods was evaluated by calculating the minimum relative error (Min(MRE)) and the minimum average relative error (Min(MARE)) at optimal configurations, with results plotted against changes in the number of truncated modes (as depicted in Figure 12 and Figure 13). Analysis of these figures reveals that the reconstruction accuracy and stability of the POD-MLP and POD-RBF methods demonstrate a non-monotonic trend, initially increasing and then decreasing. This suggests that both methods struggle to maintain robust predictive performance at low (<6) or excessively high (>14) mode counts. Conversely, the Gappy POD and Gappy C-POD methods display a rapid enhancement in reconstruction performance with an increasing number of modes, eventually stabilizing, which indicates their lower sensitivity to mode truncation and greater robustness. Notably, within the range of 6 to 11 modes, all four methods exhibit commendable reconstruction accuracy, with the Gappy C-POD method showcasing exceptional stability in this interval, maintaining a stable error level even as the number of modes rises. In contrast, the POD-RBF method achieves its best reconstruction performance at 6 modes but experiences significant fluctuations in performance, particularly when the mode count is low (≤4) or high (>14), resulting in a marked increase in error and a substantial drop in stability.

The influence of modal number on reconstruction performance was further quantified by comparing the maximum relative error (MRE) and mean absolute relative error (MARE) across different truncation levels. Specifically, the Gappy C-POD method maintained high stability in the range of 6 to 11 modes, with MRE fluctuating narrowly between 0.0378 and 0.0405, and MARE remaining below 0.0056. Gappy POD showed similar stability, though its errors increased slightly above 10 modes. In contrast, POD-MLP and POD-RBF exhibited greater sensitivity: for example, POD-RBF’s MRE exceeded 0.08 at six modes and only dropped below 0.075 when the number of modes reached nine or more. POD-MLP achieved its best results near 9–10 modes but degraded rapidly outside this range. These comparisons indicate that Gappy C-POD provides more robust reconstruction performance, with lower dependency on specific mode numbers.

Based on the performance data illustrated in Figure 12 and Figure 13, this study classifies the range of modal counts into three levels: low modes (≤6), moderate modes (>6 and ≤10), and high modes (>10). The stability and overall reconstruction accuracy of the four methods across these intervals have been qualitatively ranked as follows:
(1)Stability Ranking:
Low Modes: Gappy POD > Gappy C-POD > POD-MLP > POD-RBF;Moderate Modes: Gappy C-POD > Gappy POD > POD-MLP > POD-RBF;High Modes: Gappy C-POD > Gappy POD > POD-MLP > POD-RBF.
(2)Overall Accuracy Ranking:
Low Modes: Gappy POD > POD-RBF > POD-MLP > Gappy C-POD;Moderate Modes: Gappy POD > Gappy C-POD > POD-MLP > POD-RBF;High Modes: Gappy C-POD > Gappy POD > POD-MLP > POD-RBF.


### 5.3. The Impact of Dataset Scale on Reconstruction Capability

To investigate the influence of varying dataset scales on the reconstruction accuracy of transient thermal temperature fields, we examined dataset sizes of 5, 10, 20, 30, 40, and 50 times the number of independent variables. The changes in the Min(MRE) and Min(MARE) values for the four different methods across these dataset scales are illustrated in Figure 14 and Figure 15. The results presented in these figures reveal a clear trend: as the dataset size increases, both the precision and stability of the reconstruction methods improve progressively for transient thermal problems.

Quantitatively, the Gappy C-POD method achieved a Min(MARE) below 0.006 when the dataset size reached 30× the number of independent variables and showed marginal improvements beyond 40×, indicating convergence. Gappy POD demonstrated similar convergence behavior, stabilizing around 0.005 in MARE at 40× scale. In contrast, POD-MLP continued to improve beyond 40× scale but started with significantly higher errors (MARE > 0.01) at small dataset sizes (e.g., 5× or 10×). POD-RBF showed unstable performance for small-scale datasets, with fluctuating MRE values above 0.08, and only achieved acceptable accuracy (MRE < 0.05) when dataset size exceeded 40×. These trends suggest that Gappy C-POD provides more efficient learning with limited data and reaches stable accuracy faster than the other methods.

### 5.4. Discussion of Optimal Reconstruction Strategies for Transient Thermal Problems

In conjunction with the discussion in Section 3.2 regarding the number of modes, a ninth-order mode was selected for further analysis. Additionally, as shown in Figure 14 and Figure 15, the variations in accuracy and stability become relatively stable when the dataset size exceeds 30 times the number of independent variables. Taking into account computational and storage efficiency, a dataset scale of 30 times the number of independent variables was chosen. Under these conditions, we analyzed the reconstruction strategies that yield optimal results for each method, as summarized in Table 8 and Table 9.

Table 8 indicates that, with respect to the Min(MRE) metric, the Gappy C-POD method, when combined with the CCFM (Cross-Correlation Filtering Method) for sensor selection and a configuration of 16 sensors, demonstrates the strongest capability for minimizing reconstruction error, achieving a Min(MRE) value of 0.0371, which significantly outperforms the other three methods. The Gappy POD method, utilizing S-OPT for sensor selection with the same number of sensors, also shows stable performance with a Min(MRE) of 0.0462, indicating its effective error-control capabilities. In contrast, the POD-MLP and POD-RBF methods present minimum maximum errors of 0.0487 and 0.0561, respectively, reflecting somewhat lower overall stability due to their susceptibility to sensor selection sensitivity and the instability of nonlinear mapping.

As observed in Table 9, regarding the Min(MARE) metric, both the Gappy C-POD and Gappy POD methods exhibit strong overall precision. The former, using CCFM with nine sensors, achieves a minimum value of 0.00456, while the latter, utilizing S-OPT with nine sensors, attains a value of 0.00462, demonstrating nearly equivalent performance. This suggests that under the ninth-order mode with moderate sample density, both POD-based truncation methods offer high data utilization efficiency and precision advantages. Although the POD-MLP and POD-RBF methods yield relatively lower precision, they can still deliver results with fewer sensor configurations.

## 6. Conclusions

This study addresses the inverse reconstruction of temperature fields in two-dimensional steady-state and transient heat conduction problems. Rather than a simple comparison of reconstruction techniques, the core objectives of this work were to construct and validate a complete technical framework for the Gappy C-POD method and to systematically evaluate the interaction between sensor layout strategies and data-driven reconstruction methods within an inverse problem setting. The major conclusions are summarized as follows:

For steady-state heat conduction problems, the following was found:(1)Gappy C-POD, combined with the CCFM and maximum distance sampling, achieves the lowest maximum relative error (Min(MRE = 0.0373)) with 49 sensors and low modal numbers (1–5), demonstrating superior error-constraining ability.(2)Gappy POD shows excellent average accuracy (Min(MARE = 0.00048)) with high modal numbers (15–20), a 50 × database size, and uniform sensor placement.(3)Compared to neural network-based methods, Gappy C-POD requires fewer samples to achieve stable performance, showing higher robustness in engineering scenarios with limited data.

For transient heat conduction problems, the following was found:(1)Gappy C-POD maintains stable reconstruction under a modal range of 6–11 and achieves the lowest maximum error (Min(MRE = 0.0371)) and mean error (Min(MARE = 0.00456)) with fewer sensors (16 and 9, respectively).(2)POD-MLP and POD-RBF are more sensitive to sensor layouts and sample size. POD-MLP is more suitable for real-time applications when data is abundant and speed is prioritized.

Our overall assessment is as follows:(1)Gappy C-POD offers a balanced trade-off between reconstruction stability and computational efficiency, particularly effective for sensor-limited and low-mode scenarios.(2)Gappy POD is preferable when high-precision reconstruction is essential.(3)Neural network-based methods serve as valuable complements in data-rich, computation-permissive environments.

To consolidate the above conclusions and provide a practical overview of the methods discussed, Table 10 summarizes the advantages, limitations, and suitable application scenarios of all four reconstruction methods evaluated in this study. This comparison aims to assist future researchers and practitioners in selecting appropriate techniques for inverse temperature field problems under different constraints.

## Figures and Tables

**Figure 1 sensors-25-04984-f001:**
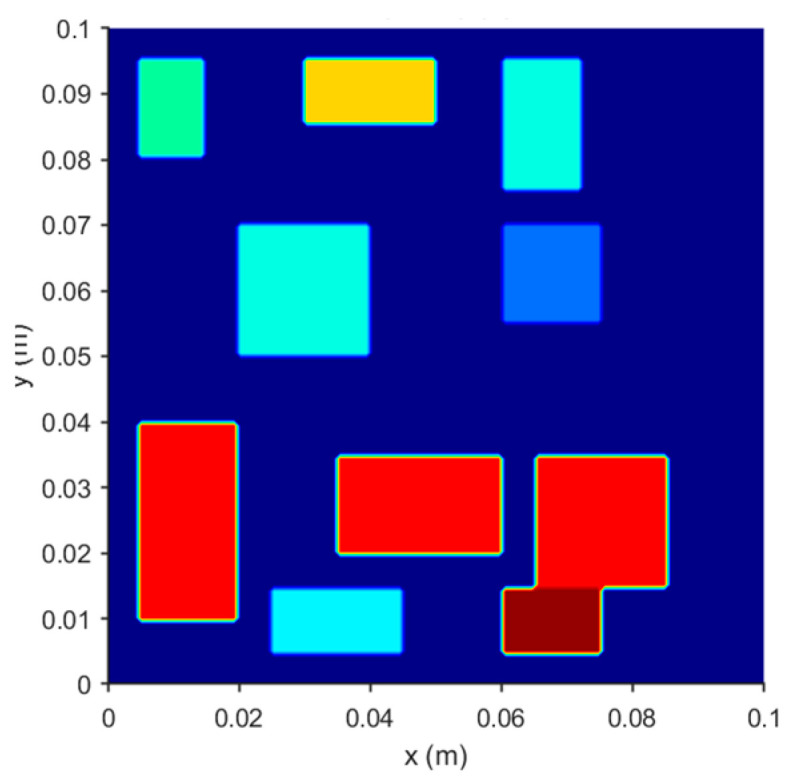
Two-dimensional steady-state heat conduction problem.

**Figure 2 sensors-25-04984-f002:**
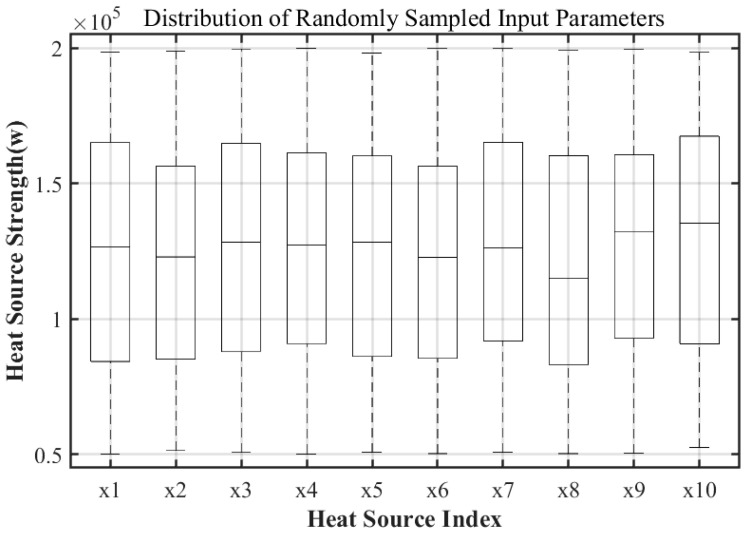
Distribution of the test set input parameters (boxplot).

**Figure 3 sensors-25-04984-f003:**
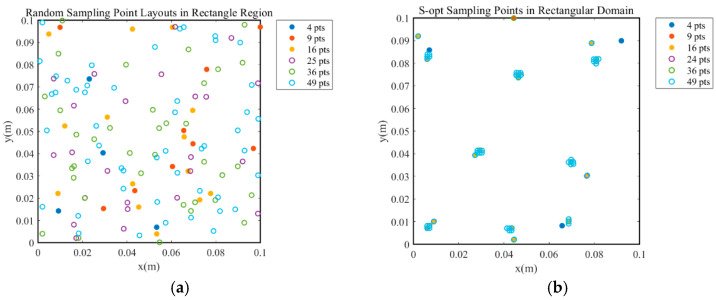
Layouts of four types of sparse sensors: (**a**) random point selection; (**b**) S-OPT method; (**c**) CCFM; (**d**) uniform point selection.

**Figure 4 sensors-25-04984-f004:**
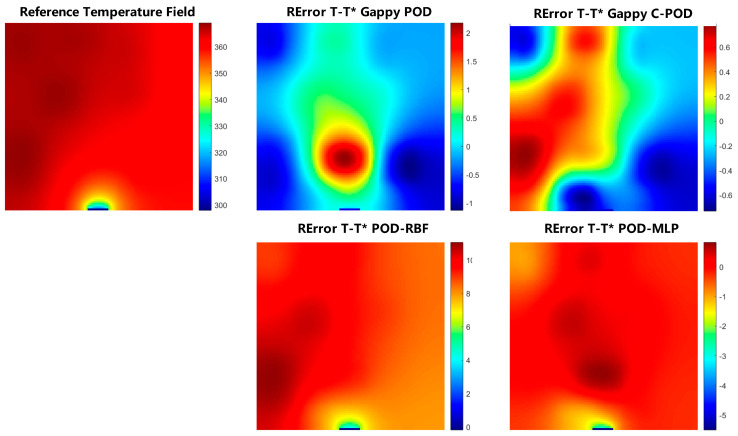
Distribution of reconstruction errors for each method on the validation set.

**Figure 5 sensors-25-04984-f005:**
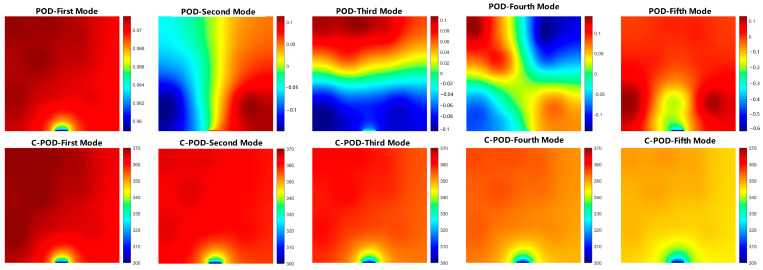
POD and C-POD reduced-order modes.

**Figure 6 sensors-25-04984-f006:**
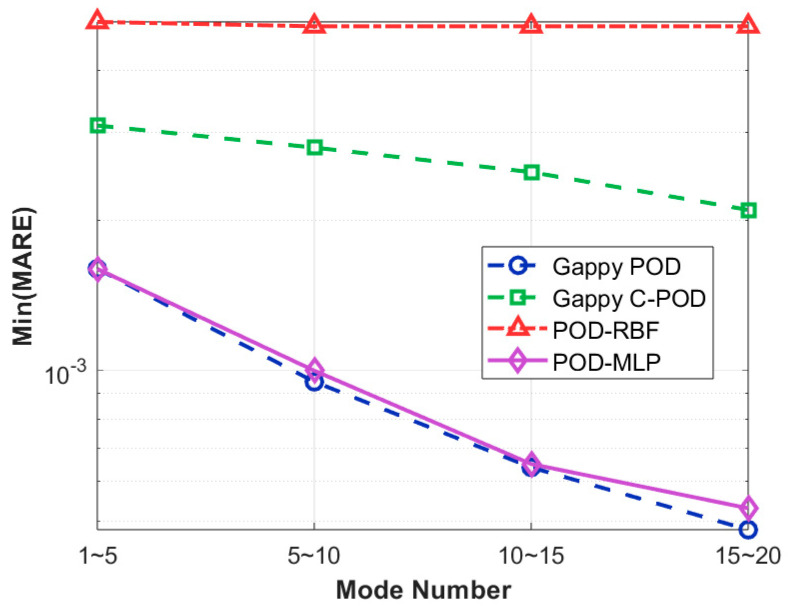
Variation in the minimum mean absolute relative error (MARE) values corresponding to different numbers of modalities.

**Figure 7 sensors-25-04984-f007:**
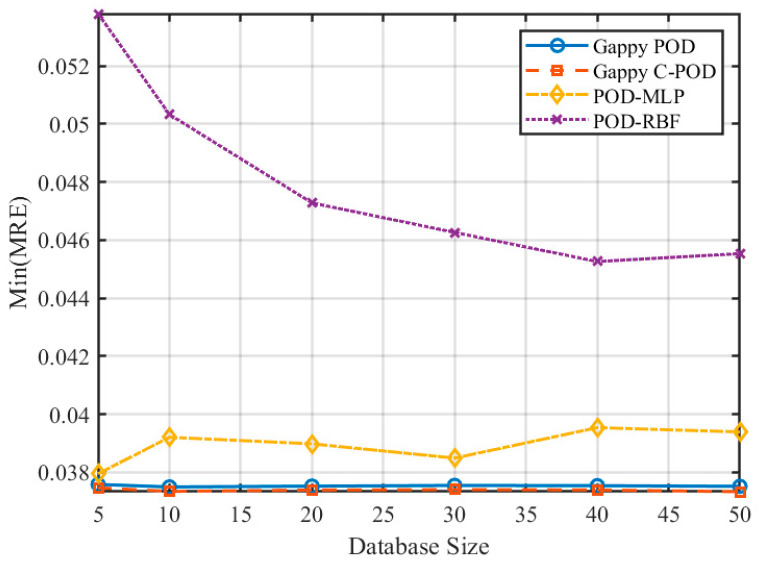
Variation in Min(MRE) for four reconstruction methods under different database scale conditions.

**Figure 8 sensors-25-04984-f008:**
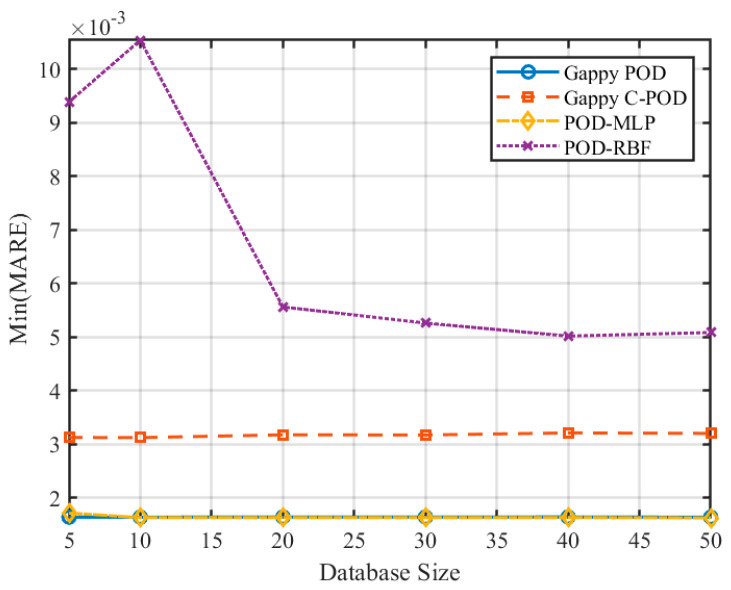
Variation in Min(MARE) for four reconstruction methods under different database scale conditions.

**Figure 9 sensors-25-04984-f009:**
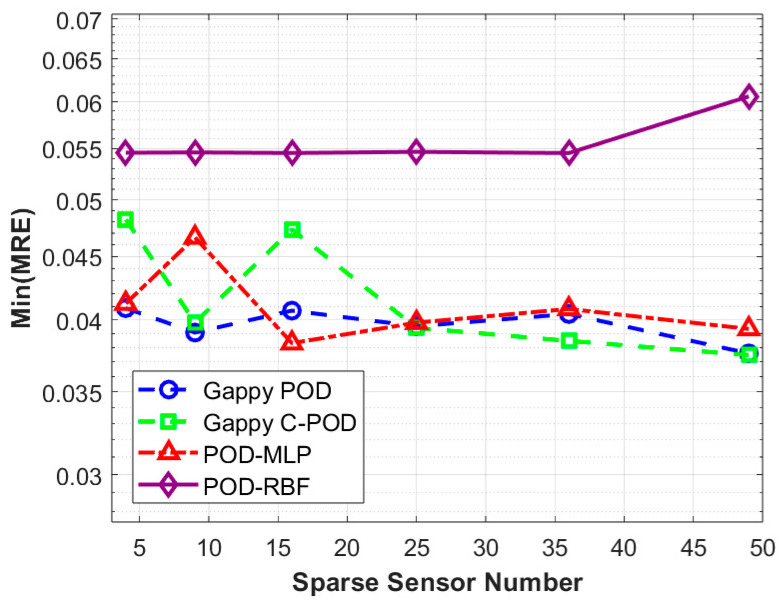
Variation in Min(MRE) values with number of sensors under four reconstruction methods.

**Figure 10 sensors-25-04984-f010:**
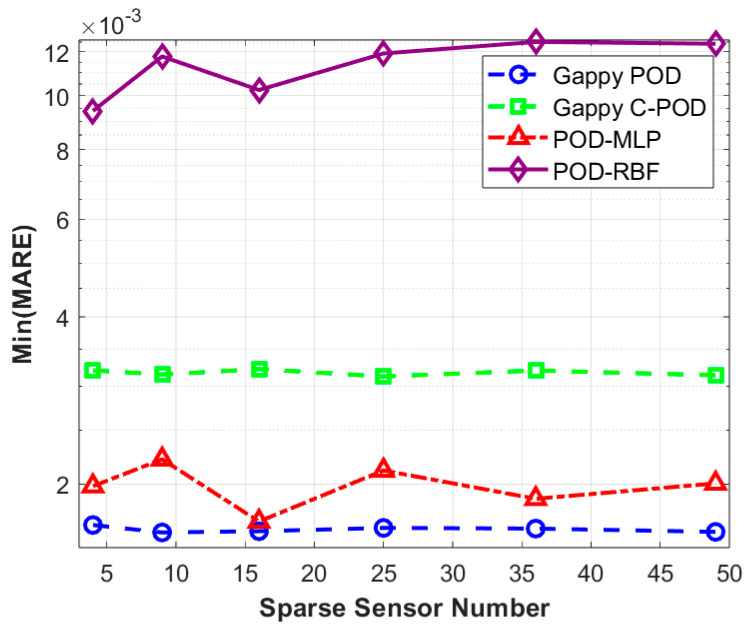
Variation in Min(MARE) values with number of sensors under four reconstruction methods.

**Figure 11 sensors-25-04984-f011:**
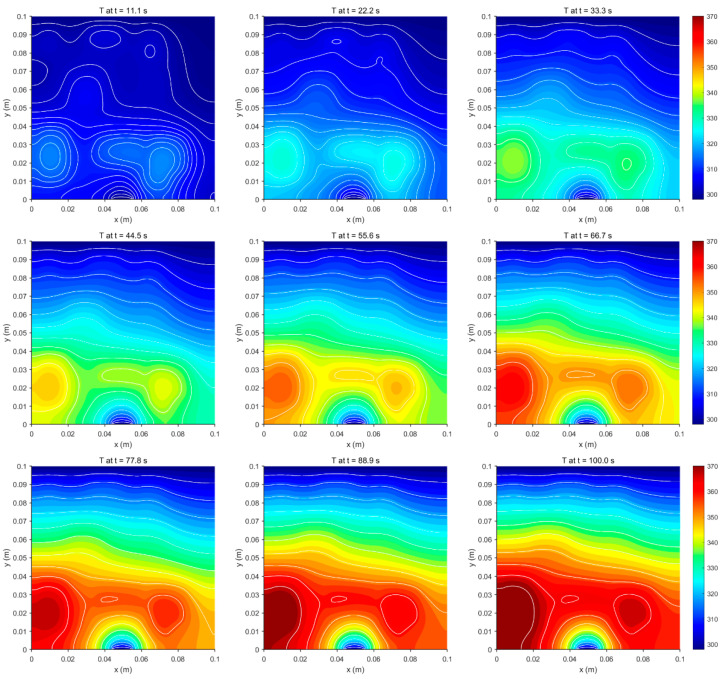
A schematic diagram of the evolution of the transient thermal conductivity temperature field distribution.

**Figure 12 sensors-25-04984-f012:**
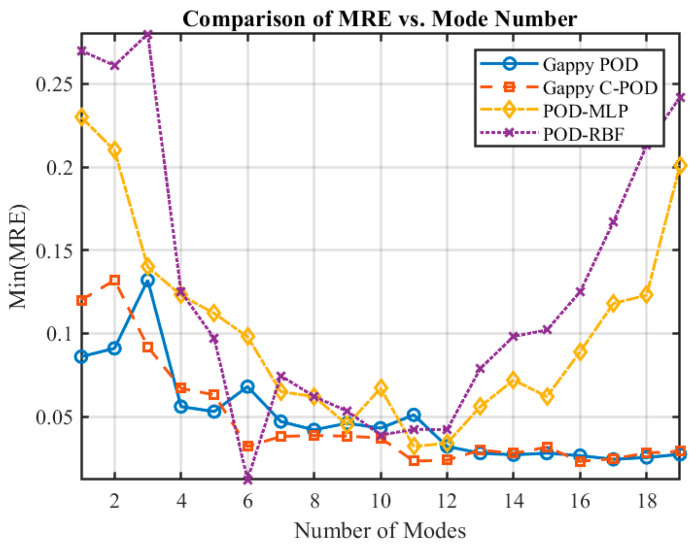
The Min(MRE) values achievable by various reconstruction methods under different conditions of sparse sensor quantities.

**Figure 13 sensors-25-04984-f013:**
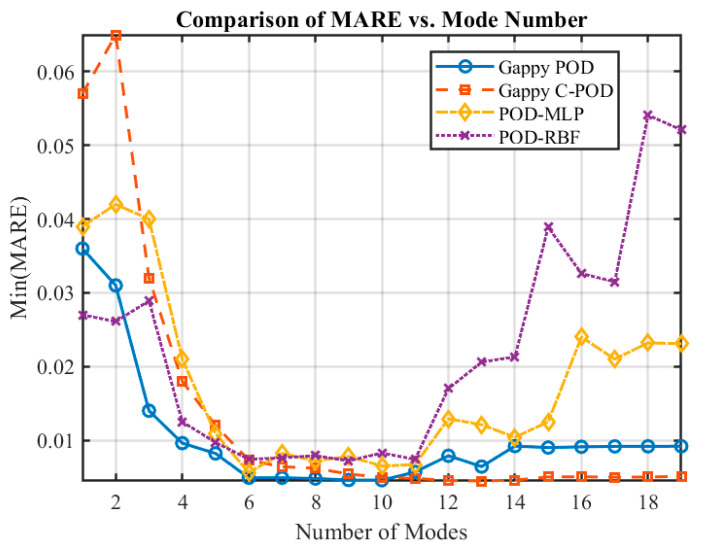
The Min(MARE) values achievable by various reconstruction methods under different conditions of sparse sensor quantities.

**Figure 14 sensors-25-04984-f014:**
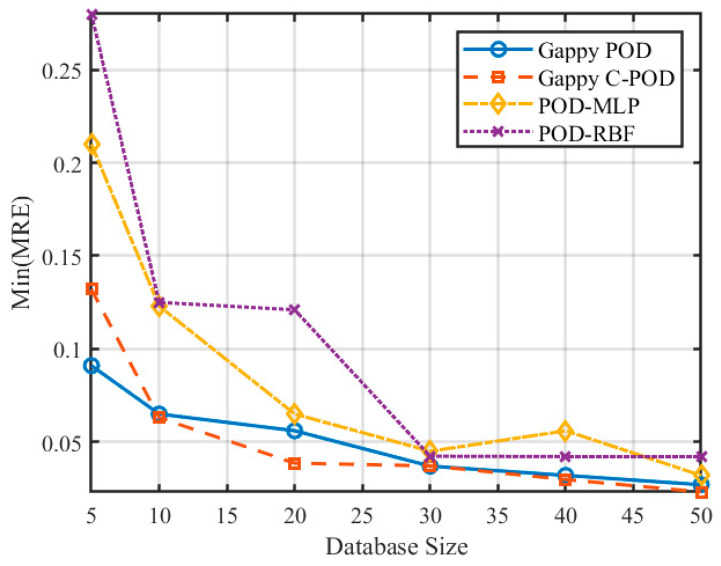
Variation in Min(MRE) values for different reconstruction methods under different dataset scale conditions.

**Figure 15 sensors-25-04984-f015:**
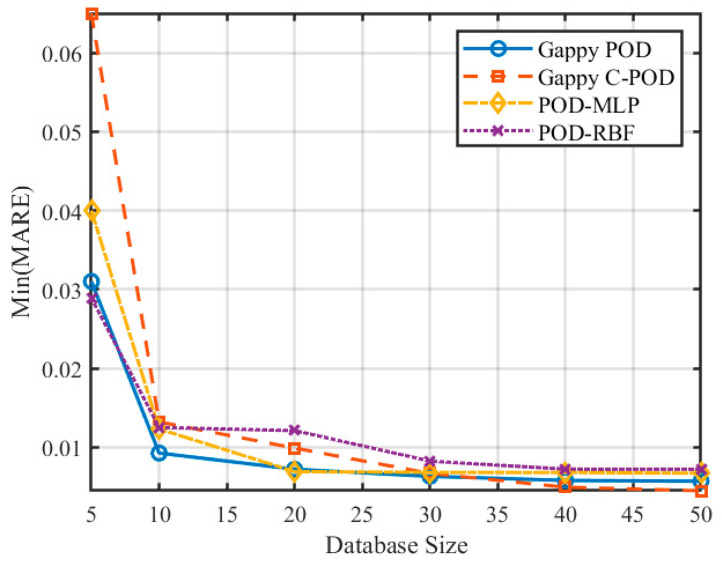
Variation in Min(MARE) values for different reconstruction methods under different dataset scale conditions.

**Table 1 sensors-25-04984-t001:** Properties and intensity parameters of heat sources.

Heat Source Identification Number	Starting Position	Dimensions	Intensity ϕ (W/m^2^)
1	(0.005, 0.005)	(0.010, 0.015)	0.5 × 10^5^~2.0 × 10^5^
2	(0.030, 0.005)	(0.020, 0.010)	
3	(0.060, 0.005)	(0.012, 0.020)	
4	(0.020, 0.030)	(0.020, 0.020)	
5	(0.060, 0.030)	(0.015, 0.015)	
6	(0.005, 0.060)	(0.015, 0.030)	
7	(0.035, 0.065)	(0.025, 0.015)	
8	(0.065, 0.065)	(0.020, 0.020)	
9	(0.025, 0.085)	(0.020, 0.010)	
10	(0.060, 0.085)	(0.015, 0.010)	

**Table 2 sensors-25-04984-t002:** Minimum MRE values achieved by various reconstruction methods under low-order modal conditions.

Implementation Strategies	Min(MRE)
Gappy POD_Maximum_10_CCFM_49	0.0375
Gappy C-POD_Maximum_50_CCFM_49	0.0373
POD-RBF_Sobol_50_CCFM_25	0.0455
POD-MLP_Sobol_20_CCFM_9	0.039

**Table 3 sensors-25-04984-t003:** Approaches for achieving minimum MARE values with various reconstruction methods under high-order modal conditions.

Implementation Strategies	Min(MARE)
Gappy POD_Maximum_50_Uniform_9	0.00048
Gappy C-POD_Maximum_5_CCFM_25	0.0021
POD-RBF_Sobol_50_CCFM_25	0.0049
POD-MLP_Maximum_50_SOPT_49	0.00053

**Table 4 sensors-25-04984-t004:** Changes in Min(MRE) and Min(MARE) values with varying numbers of sparse sensors.

Number of Sparse Sensors	Gappy POD		Gappy C-POD		POD-MLP		POD-RBF	
	Min(MRE)	Min(MARE)	Min(MRE)	Min(MARE)	Min(MRE)	Min(MARE)	Min(MRE)	Min(MARE)
4	0.0408	0.0016	0.0482	0.0032	0.0412	0.0019	0.0545	0.0093
9	0.0390	0.0016	0.0398	0.0031	0.0466	0.0022	0.0546	0.0117
16	0.0406	0.0016	0.0473	0.0032	0.0383	0.0017	0.0545	0.0102
25	0.0395	0.0016	0.0394	0.0031	0.0397	0.0021	0.0546	0.0119
36	0.0404	0.0016	0.0384	0.0032	0.040	0.0018	0.0545	0.0125
49	0.0375	0.0016	0.0374	0.0031	0.039	0.0020	0.0606	0.0124

**Table 5 sensors-25-04984-t005:** Sampling methods and layout methods used by each approach when achieving Min(MRE) under different sensor quantity conditions.

Number of Sparse Sensors	Gappy POD		Gappy C-POD		POD-MLP		POD-RBF	
	Sampling Method	Layout Method	Sampling Method	Layout Method	Sampling Method	Layout Method	Sampling Method	Layout Method
4	Sobol	Uniform	Latin	CCFM	Latin	CCFM	Latin	SOPT
9	Latin	SOPT	Sobol	SOPT	Sobol	Uniform	Latin	CCFM
16	Sobol	SOPT	Latin	CCFM	Latin	Uniform	Maximum	CCFM
25	Latin	SOPT	Latin	SOPT	Maximum	Uniform	Latin	CCFM
36	Sobol	Uniform	Latin	CCFM	Maximum	Uniform	Latin	SOPT
49	Latin	CCFM	Latin	CCFM	Sobol	SOPT	Latin	CCFM

**Table 6 sensors-25-04984-t006:** Sampling methods and layout methods used by each approach when achieving Min(MARE) under different sensor quantity conditions.

Number of Sparse Sensors	Gappy POD		Gappy C-POD		POD-MLP		POD-RBF	
	Sampling Method	Layout Method	Sampling Method	Layout Method	Sampling Method	Layout Method	Sampling Method	Layout Method
4	Maximum	SOPT	Sobol	SOPT	Maximum	SOPT	Maximum	CCFM
9	Sobol	Uniform	Sobol	Uniform	Sobol	Uniform	Latin	CCFM
16	Sobol	Uniform	Sobol	Uniform	Maximum	SOPT	Sobol	CCFM
25	Maximum	Random	Sobol	Uniform	Maximum	SOPT	Latin	CCFM
36	Sobol	Uniform	Sobol	Uniform	Sobol	SOPT	Sobol	CCFM
49	Sobol	Uniform	Sobol	Uniform	Sobol	SOPT	Latin	CCFM

**Table 7 sensors-25-04984-t007:** Materials and numerical parameters.

Parameter	Symbol	Value
Thermal conductivity	*k*	1
Density	*ρ*	500
Specific heat capacity	*Cp*	100
Thermal diffusivity	*α*	2 × 10^−5^
Grid size	*dx*, *dy*	1 mm
Time step	Δ*t*	0.01
Termination time	tend	100

**Table 8 sensors-25-04984-t008:** Achieving Min(MRE) under different reconstruction methods, point selection methods, and point selection quantities.

Reconstruction Methods	Selection Method	Number	Min(MRE)
Gappy POD	S-OPT	16	0.0462
Gappy C-POD	CCFM	16	0.0371
POD-MLP	Uniform	9	0.0487
POD-RBF	CCFM	9	0.0561

**Table 9 sensors-25-04984-t009:** Achieving Min(MARE) under different reconstruction methods, point selection methods, and point quantity conditions.

Reconstruction Methods	Selection Method	Number	Min(MRE)
Gappy POD	S-OPT	9	0.00462
Gappy C-POD	CCFM	9	0.00456
POD-MLP	S-OPT	4	0.00726
POD-RBF	CCFM	4	0.00823

**Table 10 sensors-25-04984-t010:** Comparative summary of four data-driven methods for inverse temperature field reconstruction.

Methods	Advantages	Limitations	Best Suited For
Gappy POD	Simple implementation; fast computation	Sensitive to sensor layout; may underperform for nonlinear fields	Moderate-sized problems with moderate nonlinearity
Gappy C-POD	Robust under sparse sampling; good generalization in low-mode conditions	More complex workflow; needs clustering + interpolation steps	Sparse sensor scenarios; low-mode truncation problems
POD-MLP	Fast convergence; good with small datasets	Sensitive to interpolation nodes; degrades with noisy input	Small-scale problems with smooth field variations
POD-RBF	Strong nonlinearity fitting; flexible network tuning	Risk of overfitting; needs large database; longer training	Highly nonlinear systems with sufficient training data

## Data Availability

The original contributions presented in this study are included in the article. Further inquiries can be directed to the corresponding author.

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
