# Peer review of "Optimization of Sparse Sensor Layouts and Data-Driven Reconstruction Methods for Steady-State and Transient Thermal Field Inverse Problems"

_sensors, 2025, doi:10.3390/s25164984_

Round 1
Reviewer 1 Report
Comments and Suggestions for Authors
The author studied four data-driven construction methods in order to provide guidance for the inverse problem of temperature field. However, there are many issues in the manuscript that need to be solved. The purpose and innovation of this study cannot be clearly defined according to the introduction. There are many related methods described but the key points are not clear. The result analysis is too simple, and the basic quantitative indicators that affect the changes are not compared. The conclusion needs to be further sorted out.
Other issues are as follows:
1. Line 16-24, it is recommended to delete some specific model introductions, supplement the description of experimental results, and quantify the parameter changes instead of just describing them.
2. It is recommended to reorganize the keywords, which should be related to the core content of this manuscript.
3. What is the innovation of this manuscript?
4. Based on the Introduction content provided in this manuscript, it is difficult to sort out the purpose of the author's research? Is it just to compare the four methods of Gappy POD, Gappy C-POD, POD-RBF and POD-MLP?
5. It is recommended to split the content of 2.3 into 2.3POD-RBF and 2.4POD-MLP, and do not describe them together.
6. You can make a table to systematically compare these methods.
7. Figure 3-5, the pictures are not clear and difficult to judge.
8. The discussion of the results of reconstructing the transient thermal conductivity temperature field (5.1, 5.2, 5.3) should have a specific parameter change (quantitative) analysis, rather than a simple content description.
9. Regarding the four data-driven reconstruction methods, which ones are newly obtained in this study? Which ones are known and need to be clarified.
10. Line 681-709, the relevant content should be reorganized to describe the most important content obtained in this study.
Author Response
The responses to the reviewers’ comments are provided in the attachment.

Reviewer 2 Report
Comments and Suggestions for Authors
The authors have presented introduction and all the methods in great detail. I have a few minor suggestions to improve the manuscript:
1) Line 344-345: MARE should mean absolute relative error.
2) The quality of Figure 5 is very poor. It is not readable at all. Please replace by better quality image.
3) Gappy C-POD seems to be the best method in terms of reconstruction error. Can the authors also comment on whether the numerical difference between these methods is also practically meaningful. In other words, what is the maximum allowable error in practical implementation of any of these methods for engineering applications ? Authors can pick an example and elaborate on practical utility of Gappy C-POD method as compared to other methods.
Author Response

(The authors gave the same response as above.)

Round 2
Reviewer 1 Report
Comments and Suggestions for Authors
The revised manuscript can be accepted.